# TreeReward: Improve Diffusion Model via Tree-Structured Feedback Learning

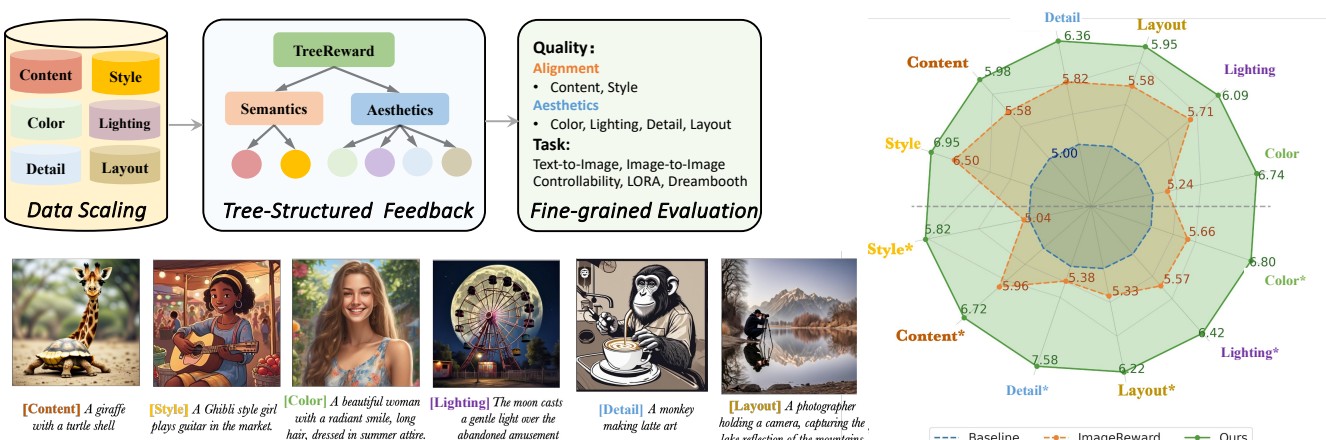

**Figure 1: We propose TreeReward, an innovative feedback learning framework that encompasses three two components: feedback data scaling up, and tree-structured feedback learning. TreeReward aims to enhance diffusion models from various aspects including Content, Style, Color, Lighting, Detail, and Layout. The radar chart visually represents the improvement achieved by our model compared to the baseline (SD1.5). In this chart, the baseline value is set at 5, and the value range is rescaled from [-5, 5] to [0, 10] for better visualization. * means that SD1.5 that fine-tuned in JourneyDB.**

## ABSTRACT

Recently, there has been significant progress in leveraging human feedback to enhance image generation, leading to the emergence of a rapidly evolving research area. However, current work faces several critical challenges: i) insufficient data quantity; and ii) rough feedback learning; To tackle these challenges, we present **TreeReward**, a novel multi-dimensional, fine-grained, and adaptive feedback learning framework that aims to improve both the semantic and aesthetic aspects of diffusion models. Specifically, To address the limitation of the fine-grained feedback data, we first design an efficient feedback data construction pipeline in an "AI + Expert" fashion, yielding about 2.2M high-quality feedback dataset encompassing six fine-grained dimensions. Built upon this, we introduce a tree-structure reward model to exploit the fine-grained feedback data efficiently and provide tailored optimization during feedback learning. Extensive experiments on both Stable Diffusion v1.5 (SD1.5) and Stable Diffusion XL (SDXL) demonstrate the effectiveness of our method in enhancing the general and fine-grained

generation performance and the generalizability of downstream tasks.

## CCS CONCEPTS

• **Computing methodologies → Computer vision**.

## KEYWORDS

Diffusion Model, Feedback Learning, Tree-structured Reward

## 1 INTRODUCTION

Reinforcement learning from human feedback (RLHF) [1, 18, 19] has recently made significant strides in enhancing large language models (LLM) [26, 32], attracting substantial interest in the field. This technique, which incorporates human feedback, aims to improve the quality and safety of outputs from language models. Concurrently, similar methodologies are being adopted for diffusion-based image generation. In these applications [29, 30], diffusion models are fine-tuned using human preference ratings, with reward functions designed to align the generated images more closely with human preferences. However, despite these developments, diffusion models that integrate human feedback learning still face several ongoing challenges. (i) Insufficient Feedback Data: Existing methods suffer from the limited preference data volume, especially the fine-grained preference feedback data, which may not sufficiently capture the diverse range of human preferences. Nonetheless, gathering large amounts of preference data is labor-intensive and costly. (ii) Coarse Feedback Learning: Due to the scarcity of tailored fine-grained feedback data, the majority of existing datasets primarily

focus on coarse feedback learning and fail to explore efficient methods for leveraging fine-grained preference fine-tuning to enhance the performance of diffusion. To address these challenges, we propose an effective fine-grained feedback-learning method to boost the performance of the diffusion model comprehensively. Specifically, to tackle the issue of lack of fine-grained feedback data, we first design an efficient feedback data construction pipeline for six fine-grained dimensions. According to the different characteristics of distinct fine-grained dimensions, it incorporated both the automatic feedback data generation and the human preference annotation to enable feedback data scale-up in a low-cost manner. With this pipeline, we scale up the feedback dataset to about 2.2M, which is the largest available dataset in the field of text-to-image generation. Based on this dataset, we further introduce a novel tree-structured reward model, namely **TreeReward**. It organizes the fine-grained feedback dimension hierarchically and utilizes the random sample ensemble training strategy to effectively integrate the scoring abilities of multiple fine-grained feedback into a single reward model. During reward feedback learning, it aggregates the reward scores from all the leaf nodes in an adaptive manner, offering the case-tailored feedback signal for optimization. Extensive experiments demonstrate the superiority of our method in enhancing the generation performance of both general quality and fine-grained dimensions. Furthermore, we validate the model's performance in downstream tasks, demonstrating its robustness and generalization. Our contributions are summarized as follows:

- We design an efficient feedback data curation pipeline that combines automated feedback generation and manual annotation to efficiently collect fine-grained feedback data on various aspects such as style, content, light, structure, layout, and detail, finally yielding about 2.2M feedback dataset which is the largest dataset in the field.
- We introduce an innovative feedback learning method based on a tree-structure reward, TreeReward, that enables multi-dimensional, fine-grained, and adaptive preference feedback learning, resulting in more comprehensive and effective reward tuning.
- Extensive experiments on both SD1.5 and SDXL validate the effectiveness of our method, showcasing superior performance when compared to the state-of-the-art reward tuning method. Additionally, the experiment on the downstream tasks further validates the efficacy and generalization of our method.

## 2 RELATED WORKS

### 2.1 Text-to-Image Generation

Text-to-image generative models, including auto-regressive [4, 6, 22], GANs [7, 11, 12] and diffusion models [8, 17, 25, 33], have become a prominent research area in various applications, attracting significant attention. Among the evolving methodologies, diffusion models have emerged as the de facto mainstream technique for text-to-image synthesis due to their impressive generation capabilities. The widespread adoption of diffusion models can be attributed to their demonstrated effectiveness, as evidenced by pioneering works [3, 9, 15, 17, 21, 23] such as DALLE-2 [21] and Stable Diffusion [23]. However, despite diffusion models having great success in

text-to-image synthesis, they still struggle to generate images that are well-aligned with the user preference within the text prompts. This paper addresses this limitation by directly incorporating fine-grained human feedback into the training of diffusion models.

### 2.2 Learning from Human Feedback

Due to the inherent noise of the pre-trained dataset, there is often a gap between generative models' pre-training objectives and human intent. To mitigate this gap, human feedback learning [1, 5, 18, 19, 31] has been utilized to align model performance with human preference in the language domain. Inspired by these works, several works have endeavored to incorporate human feedback into the learning process of diffusion models to better understand human preferences. DDPO [2] adopts a reinforcement learning framework to align diffusion model generation with the supervision provided by an additional reward model. Approaches like HPS [28, 29] employ a separate reward model trained on curated human preference datasets to filter eligible preferred data for fine-tuning stable diffusion. Another approach, Reward Weighting [14], utilizes reward-weighted likelihood as the optimization objective. ImageReward [30] proposes the ReFL training framework to fine-tune stable diffusion via a differentiable reward model. While these methods have shown success in human preference learning, they rely on a general reward model trained on coarse human preference datasets, limiting their ability to provide fine-grained preference evaluation. Additionally, they require training an extra reward model when preferences change. In this paper, we address these limitations by curating a fine-grained human preference dataset. Building upon this dataset, we propose a tree-structured reward model that offers more effective and flexible reward supervision for diffusion models.

## 3 PRELIMINARY

### 3.1 Text-to-Image Diffusion Model

Text-to-image models use diffusion modeling to create high-quality images from text prompts. The diffusion model produces data samples from Gaussian noise by gradually denoising process. During pre-training, a sampled image $x$ is first processed by a pre-trained VAE encoder to derive its latent representation $z$. Subsequently, random noise is injected into the latent representation through a forward diffusion process, following a predefined schedule $\{\beta_t\}^T$. This process can be formulated as $z_t = \sqrt{\overline{\alpha}_t}z + \sqrt{1 - \overline{\alpha}_t}\epsilon$, where $\epsilon \in \mathcal{N}(0,1)$ is the random noise with identical dimension to $z$, $\overline{\alpha}_t = \prod_{s=1}^{t} \alpha_s$ and $\alpha_t = 1 - \beta_t$. To achieve the denoising process, a UNet $\epsilon_\theta$ is trained to predict the added noise in the forward diffusion process, conditioned on the noised latent and the text prompt $c$. Formally, the optimization objective of the UNet is:

$$\mathcal{L}(\theta) = \mathbb{E}_{z,\epsilon,c,t}[||\epsilon - \epsilon_\theta(\sqrt{\overline{\alpha}_t}z + \sqrt{1 - \overline{\alpha}_t}\epsilon, c, t)||_2^2]. \quad (1)$$

### 3.2 Reward Feedback Learning

Reward feedback learning(ReFL) [30] is a preference fine-tuning framework that aims to improve the diffusion model via human preference feedback. It primarily includes two phases: (1) Reward Model Training and (2) Preference Fine-tuning. In the Reward Model Training phase, human preference data is collected. These data are then utilized to train a human preference reward model, which serves

| Name | Annotator | Data Format | Prompt Source | Image Source | # Dim. | # Prompt | # Pairs |
|------|-----------|-------------|---------------|--------------|--------|----------|---------|
| HPS [29] | Discord users | Top-1 choice | Discord users | Stable Diffusion | 1 | 25k | 25k |
| ImageReward [30] | Expert | Pairwise | DiffusionDB | DiffusionDB | 1 | 9k | 137k |
| PickScore [13] | Web users | Pairwise | Web users | 4 Models | 1 | 38k | 584k |
| HPSv2 [28] | Expert | Pairwise | DiffusionDB* | 9 Models + real photo | 1 | 108k | 798k |
| Ours | AI + Expert | Pairwise | DiffusionDB* + Web users | 15 Models + real photo | 6 | 400k | 2.2M |

**Table 1: Comparison with other datasets. * indicates that the data has been filtered. Our data covers 6 fine-grained dimensions. Compared with the previous largest dataset, our preferred pair data pair has been expanded by 2.75 ×.**

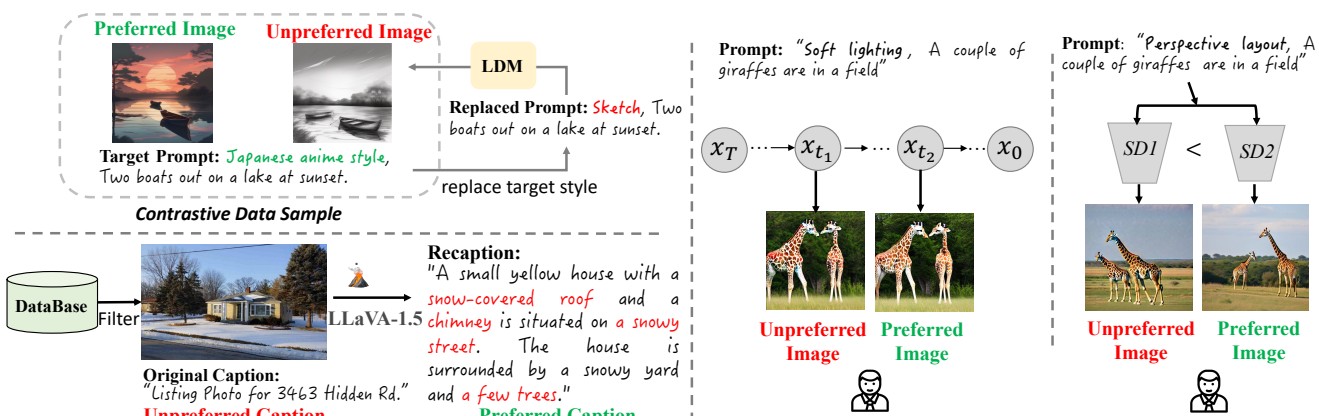

**Figure 2: The overview of our "AI + Expert" fine-grained feedback data construction pipeline. For the feedback data on different fine-grained dimensions, we design distinct strategies to generate the feedback data accordingly, leading to efficient feedback data scaling up.**

as an encoding mechanism for capturing human preferences. More specifically, considering two candidate generations, denoted as $x_w$ (preferred generation) and $x_l$ (unpreferred one), the loss function for training the human preference reward model $r_\theta$ can be formulated as follows:

$$\mathcal{L}(\theta)_{rm} = -\mathbb{E}_{(c,x_w,x_l)\sim\mathcal{D}}\left[log(\sigma(r_\theta(c, x_w) - r_\theta(c, x_l)))\right], \quad (2)$$

where $\mathcal{D}$ denotes the collected feedback data, $\sigma(\cdot)$ represents the sigmoid function, and $c$ corresponds to the text prompt. The reward model $r_\theta$ is optimized to produce a preference-aligned score that aligns with human preferences. In the Preference Fine-tuning phase, ReFL begins with an input prompt $c$, initializing a latent variable $x_T$ at random. The latent variable is then progressively denoised until reaching a randomly selected timestep $t$. At this point, the denoised image $x_0'$ is directly predicted from $x_t$. The reward model obtained from the previous phase is applied to this denoised image, generating the expected preference score $r_\theta(c, x_0')$. This preference score enables the fine-tuning of the diffusion model to align more closely with human preferences:

$$\mathcal{L}(\theta)_{refl} = \mathbb{E}_{c\sim p(c)}\mathbb{E}_{x_0'\sim p(x_0'|c)}\left[-r(x_0', c)\right]. \quad (3)$$

## 4 EFFICIENT FEEDBACK DATA SCALING

An essential challenge of feedback learning resides in the collection and construction of a high-quality dataset of human feedback. Although there are already some available feedback datasets, such as the ImageReward[30] and Pickascore[13], these datasets often suffer from coarse feedback annotation and limited data volume.

However, collecting large amounts of fine-grained human feedback data is time-consuming and expensive. To tackle this issue, we design an efficient pipeline to reduce the cost of feedback data construction via a series of automatic strategies. The core insight of our method is *not all feedback data on each aspect requires accurate human annotation and we can combine the automatic data generation with manual human annotation to reduce the cost of the data construction*. The comparison between our collected feedback data and the existing dataset is summarized in Tab.1. In total, we collect about 2M feedback data across six dimensions encompassing both **semantic** and **aesthetic** aspects with such pipeline, which will be elaborated in the following sections.

### 4.1 Feedback Data on Semantics Alignment

The semantic alignment between the text prompt and the generated image is an important aspect of evaluating the performance of a text-to-image diffusion model. Rather than treating the semantic alignment as a whole [30], we further break down the semantic alignment into style alignment and content alignment to aid the fine-grained feedback data collection.

**Style Alignment.** To create a feedback dataset on style generation, we develop a **contrastive data sample** strategy to construct the style feedback data. As depicted in Fig.2, we initially collect a diverse set of approximately 500 commonly used target style words from the user prompts(e.g. prompt in the JourneyDB[20]). Subsequently, given a prompt containing a particular target style word, we randomly substitute the target style word with another word from the vocabulary. Then, we generated two images with the original prompt and the replaced prompt. Thus, for the target style,

the image generated by the target style word serves as a positive sample, whereas the image generated by the randomly substituted style word acts as a negative sample. To ensure the quality of the curated feedback data, we exploit the state-of-the-art diffusion model to generate images, such as SDXL and Kindmisky. By employing this methodology, we collect precise and detailed style feedback data that assist in training the diffusion model to accurately capture the desired style alignment in the generated images.

**Content Alignment.** Given the input prompt, we expect the diffusion model to generate all the entities and attributes mentioned in the input prompts accurately. However, the current text-to-image diffusion model still lags in this aspect due to the noisy pre-trained datasets like LAION. In this work, we attempt to collect a feedback dataset on content alignment and tackle this problem via feedback learning. Specifically, as depicted in Fig.2, we introduce a **recaption** strategy to curate the feedback data with two steps. i) Identifying Misaligned Examples: We utilized the clip model to identify image-text pairs in the LAION dataset where the clip score fell below a certain threshold. These pairs were considered instances of poor content alignment and sent to re-caption. ii) Generating Detailed Image Descriptions: Given these misaligned text-image pairs, instead of generating a more aligned image, we inversely re-generate a more aligned prompt via advanced multimodal large language models (MLLMs) such as LLaVA. iii) Feedback Data Construction: For each image, we considered the original caption and the regenerated caption from the MLLM as feedback data for content alignment with the regenerated caption regarded as the preferred alignment as a more detailed and accurate description of the image.

## 4.2 Feedback Data on Aesthetics Quality

The aesthetic quality is another critical aspect of generation performance. Due to the inherent abstract and subjective, it is hard for the diffusion model to grab the aesthetic essence. Therefore, the current methods propose to exploit human preference to steer the diffusion in the right direction. However, the abstract aesthetic concept contains quality in various dimensions, such as color, lighting, layout, and details, and the existing methods only consider coarse-grained aesthetics and cannot refine each dimension of aesthetics, and also has the risk of optimization conflict during feedback tuning as analyzed in [27]. To address this limitation, we propose to decouple the aesthetic into these dimensions and introduce an AI-assisted feedback data construction strategy.

**Color, Lighting, Layout, and Detail.** The primary challenge of fine-grained aesthetic feedback data curation lies in the lack of the image pair focusing on a particular aspect to annotate. The ordinary image pair generated by a text-to-image diffusion model of the same prompt tends to have a similar aesthetic quality, leading to hard judgment and requiring much more time to distinguish the reference sample. To ease the human annotation burden, we design two strategies to make the AI-assisted feedback data generation. Specifically, we utilize a text-to-image diffusion model to generate several images for a particular prompt and then ask the annotator to select the preferred and unpreferred sample. During this process, (i) To make the model to generate the image that focuses on a particular aspect, we manually curate a set of trigger words for each aesthetic dimension. For instance, in the lighting dimension, trigger

words like "Soft lighting," "Side lighting," "rim lighting," and "Moody lighting" are included. By incorporating these trigger words into the input prompt, we enhance the model's focus on specific aesthetic dimensions during the image generation. (ii) To ease the judgment process, we manually create the sample with aesthetic differences as depicted in Fig.2. On one hand, we utilize diffusion models with varying generative capabilities to generate the image pair. For example, we take the image generated by SDXL and its improved version Kindminsky as the candidate images for easier preference annotation. On the other hand, we take the images generated at different denoised timesteps to assist the fast preference annotation. With these two strategies, we can achieve efficient fine-grained aesthetic feedback data annotation, and reduce the annotation cost.

## 5 TREE-STRUCTURED FEEDBACK LEARNING

Given the collected fine-grained feedback dataset, the next question is how to utilize these datasets efficiently. The most direct way is to utilize these datasets as a whole and train a simple reward model as the practice in [30]. However, this way overlooks the difference between these dimensions, leading to inefficient feedback learning. To this end, we introduce a **TreeReward**, an effective way to combine fine-grained feedback data and dynamically provide fine-grained and adaptive preference feedback for fine-tuning diffusion models. Fig.3 illustrates the TreeReward training and preference fine-tuning process, which will be explained in the following sections.

### 5.1 TreeReward Training

In general, TreeReward exhibits a hierarchical structure comprised of two internal nodes and six local leaf nodes. The role of the internal nodes is to determine which general aspect (such as semantic or aesthetic) to reward, while the leaf nodes are responsible for providing precise reward scores along fine-grained dimensions. In line with ImageReward[30], we implement TreeReward with the BLIP model as the backbone, which employs ViT-L14 as the image encoder and a 12-layer transformer as the text encoder. For the internal node and leaf node of our TreeReward, we implement them with simple 2x and 4x MLPs, respectively.

We take a **random sample ensemble** strategy to train our TreeReward. Specifically, we first randomly sampled a data point from the collected feedback dataset, which contains the reward dimension, reward data. Then, a path is chosen from the root of TreeReward to the leaf node according to the reward dimension. Formally, for the internal nodes,

$$\mathcal{L}_{internal}(\theta) = \text{CrossEntropy}(g, G(x)). \quad (4)$$

Here, $G(\cdot)$ represents the predicted logits of the internal node, and $g$ is the target internal reward label determined by the source of the training data. This objective aims to let the model know which aspect to reward when given a text-image pair. For the leaf node, we expect the leaf node to output the correct reward score. We take a similar way with E.q.2 to optimize the leaf node along a particular reward dimension. Specifically, for the $j$-th reward local leaf node denoted as $R_\theta^j$, the preference feedback reward data pair is represented as $(x_w, x_l)$. Here, $x_w$ corresponds to the preferred text-image pair, while $x_l$ represents the non-preferred text-image

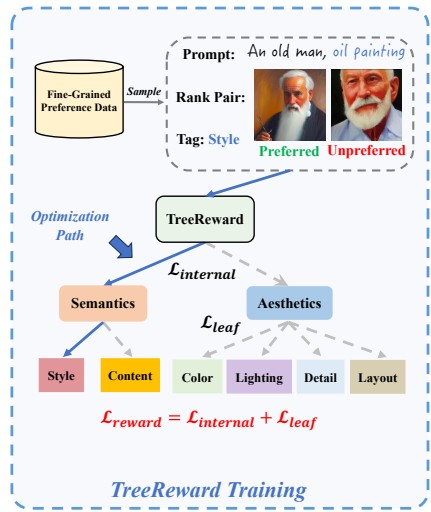
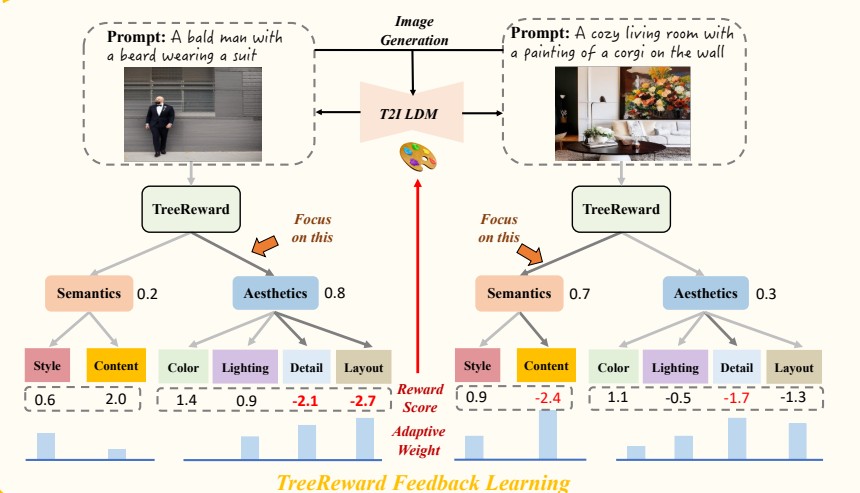

**Figure 3: The overview of our TreeReward, a tree-structured reward model trained with vast high-quality fined-grained preference data to facilitate more effective feedback learning for text-to-image generative models.**

pair. The loss function for the leaf can be formulated as:

$$\mathcal{L}_{leaf}(\theta) = -\mathbb{E}(x_w, x_l) \sim D_j [\log(\sigma(r_\theta^i(x_w) - \sigma(r_\theta^i(x_l))))]. \quad (5)$$

In the above equation, $r_\theta^j(x.)$ is the scalar reward predicted by the $j$-th leaf node, $D^j$ represents the feedback dataset for the corresponding fine-grained reward dimension of the $j$-th leaf node. The complete loss for training TreeReward is defined as:

$$\mathcal{L}(\theta)_{RM} = \mathcal{L}_{internal}(\theta) + \mathcal{L}_{leaf}(\theta). \quad (6)$$

It is worth noting that we only optimize one particular path in the TreeReward according to the source of the feedback data for each time, and leave the parameters of other nodes unchanged. However, with the random sampling training data, the whole tree reward will be fully optimized via a path ensemble way.

## 5.2 TreeReward Feedback Learning

Aligned with the practice in ImageReward [30], we adopt the direct preference fine-tuning fashion to harness the reward guidance offered by our TreeReward. However, rather than obtaining a coarse reward score, we utilize an adaptive reward score derived from our tree-structured reward model to fine-tune the diffusion model comprehensively.

Specifically, given the generated image $y_i$ and the corresponding prompt $x_i$, we calculate the reward score for the input text-image pair $(x_i, y_i)$ hierarchically and adaptively. Starting from the root node, we first calculate the internal node logits and obtain the reward weights for the semantic alignment and aesthetic quality. Formally, we have:

$$w_b = \frac{e^{r_b(x_i, y_i)}}{\sum e^{r_b(x_i, y_i)}}, \quad (7)$$

where $r_\theta^b(x)$ with $b \in \{semantic, aesthetic\}$ represents the prediction output of the internal node, and $w_b$ is the reward weight of along these two aspects. Next, we obtain all the fine-grained reward scores on the leaf node and compute the adaptive weight of each

leaf node under the internal node:

$$w_k = \frac{e^{r_k(x_i, y_i)}}{\sum e^{r_k(x_i, y_i)}}, \quad$$

where $k$ is the leaf nodes under a internal node and $w_k$ is the corresponding weight. The final reward $R$ is obtained by combining the rewards hierarchically from the root to the leaf:

$$R_{tr}(y_i, g_\theta(y_i)) = \sum_{b=0}^{M} w_b \sum_{k=0}^{N_b} w_k \cdot r_k(x_i, y_i), \quad (8)$$

where $N_b$ is the number of reward leaf nodes under the branch $b$, and $M$ is the number of global reward branches. By combining the fine-grained rewards from all nodes, our model can adaptively focus on the reward dimensions that have not been well optimized yet, providing case-tailored preference feedback for the diffusion model via:

$$\mathcal{L}_{reward} = \mathbb{E}y_i \sim y [-R_{tr}(x_i, y_i)]. \quad (9)$$

Following [30], we also incorporate the naive diffusion pre-train loss [30] as a regularization term:

$$\mathcal{L}_{pretrain} = \mathbb{E}_{(y_i, x_i) \sim D} \left( \mathbb{E}_{\mathcal{E}(x_i), y_i, \epsilon \sim \mathcal{N}(0,1), t} \left[ \|\epsilon - \epsilon_\theta(z_i, t, \tau_\theta(y_i))\|_2^2 \right] \right). \quad (10)$$

Therefore, the final training objective is:

$$\mathcal{L} = \mathcal{L}_{pretrain} + \lambda \mathcal{L}_{reward}, \quad (11)$$

where $\lambda$ is the loss weight and is set to 0.05 by default.

## 6 EXPERIMENTS

### 6.1 Implementation Details

**Dataset and Training Setting.** We utilize the collected fine-grained preference data to train our TreeReward. We randomly sample a prompt set from DiffusionDB following [30] for preference fine-tuning. We conduct experiments with Stable Diffusion v1.5 and

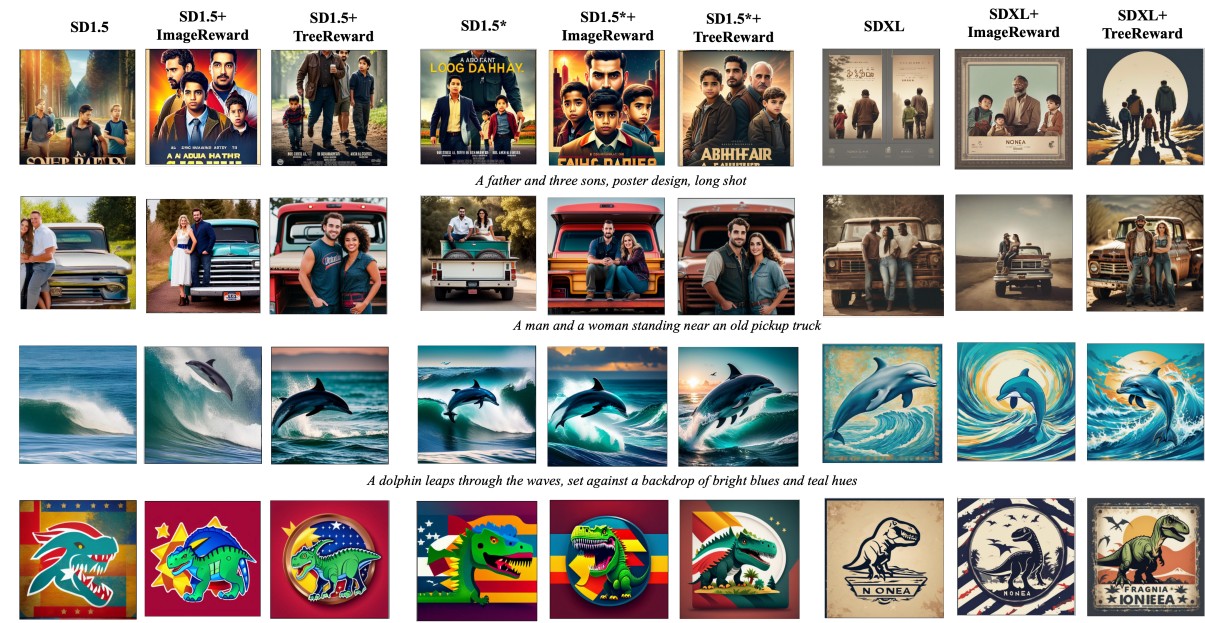

**Figure 4: Visual comparison of SOTA models. TreeReward has achieved more excellent results than other competitive methods.**

**Table 2: The quantitative results between the SOTA model and ours in clip and aesthetic score. Among them, * means that the model has been fine-tuned in JourneyDB.**

| Model | CLIP Score | Aesthetic Score |
|---|---|---|
| SD1.5 | 25.60 | 5.41 |
| SD1.5+ImageReward | 26.20 | 5.55 |
| SD1.5+TreeReward | **26.70** | **5.62** |
| SD1.5* | 26.60 | 5.89 |
| SD1.5*+ImageReward | 26.90 | 5.90 |
| SD1.5*+TreeReward | **27.40** | **5.96** |
| SDXL* | 27.28 | 5.69 |
| SDXL+ImageReward | 27.35 | 5.66 |
| SDXL+TreeReward | **27.39** | **5.84** |

Stable Diffusion XL base 1.0. Additionally, to validate the effectiveness and generalization of our method, we further utilize the JourneyDB [20], a large-scale generated image dataset collected from Midjourney, to fine-tune the base SD1.5 to acquire an improved base diffusion model where we subsequently conduct the preference fine-tuning.

**Fine-Grained Evaluation.** In addition to the general performance evaluation on the overall quality, we further constructed a fine-grained evaluation benchmark to comprehensively evaluate the model performance on the fine-grained dimensions. Specifically, we request the ChatGPT to write several typical prompts for each fine-grained dimension (i.e., content alignment, style alignment, color, lighting, detail, and layout). These prompts ensure that describe the picture that is most relevant to the corresponding fine-grained dimension. We further execute the manual check to filter out the invalid prompt. This finally results in 100 prompts for each dimension and 600 prompts in total. The examples of these prompts and details

of the evaluation procedure are displayed in the supplementary material.

## 6.2 Comparision with State-of-the-art

**Qualitative Results.** We compared our method with ImageReward, the current state-of-the-art SD preference modeling method. It clearly shows that our method exhibits a superior preference for learning performance in both semantic alignment and aesthetic quality enhancement. As shown in Fig.4, our method exhibits **superior overall visual quality**. Take SD1.5 and the prompt of "*A dolphin leaps through the waves, set against a backdrop of bright blues and teal hues*" as an example, there are no dolphins in the image generated by SD1.5. The dolphin generated by ImageReward is either too small, or the waves are blurry. By contrast, both the waves and dolphins generated by TreeReward are rich in detail and highly realistic. As depicted in Fig.5, TreeReward also shows superiority in generating images with **better visual quality in various fine-grained aspects**. For example, only TreeReward generates the correct result for prompt "*a horse without a rider*", while both the base model and ImageReward generate the mismatched content (The riders). And for prompt "*A mountain retreat's spa, zen-inspired, ... overlook forest views*", both SD1.5 and ImageReward present unreasonable layouts for the tables and swimming pool (Too small tables and truncated swimming pool), while TreeReward displays the more aesthetic layout. Note that the ImageReward does not exhibit much improvement when applied to the improved stable diffusion base model which is fine-tuned with JourneyDB. As a comparison, Our TreeReward still delivers notable improvement, which demonstrates the superiority of our method.

**Quantitative results.** To evaluate the performance of our method quantitatively, we conducted comparisons using CLIP scores and aesthetic scores, which provide metrics for semantic alignment

| Method | SD1.5 | | | | | | JourneyDB-tuned SD1.5 | | | | | |
|---|---|---|---|---|---|---|---|---|---|---|---|---|
| | Content | Style | Color | Lighting | Detail | Layout | Content | Style | Color | Lighting | Detail | Layout |
| ImageReward [30] | 0.58 | 1.50 | 0.24 | 0.71 | 0.82 | 0.58 | 0.96 | 0.40 | 0.66 | 0.57 | 0.38 | 0.33 |
| TreeReward (Ours) | 0.98 | 1.95 | 1.74 | 1.09 | 0.98 | 0.95 | 1.72 | 0.82 | 1.80 | 1.42 | 2.58 | 1.22 |

Table 3: Human fine-grained evaluation results for 2 base models optimized via 2 different reward models. It can be seen that TreeReward brings greater improvement than ImageReward in all dimensions. The value is the absolute value of improvement, and the range is [-5, 5].

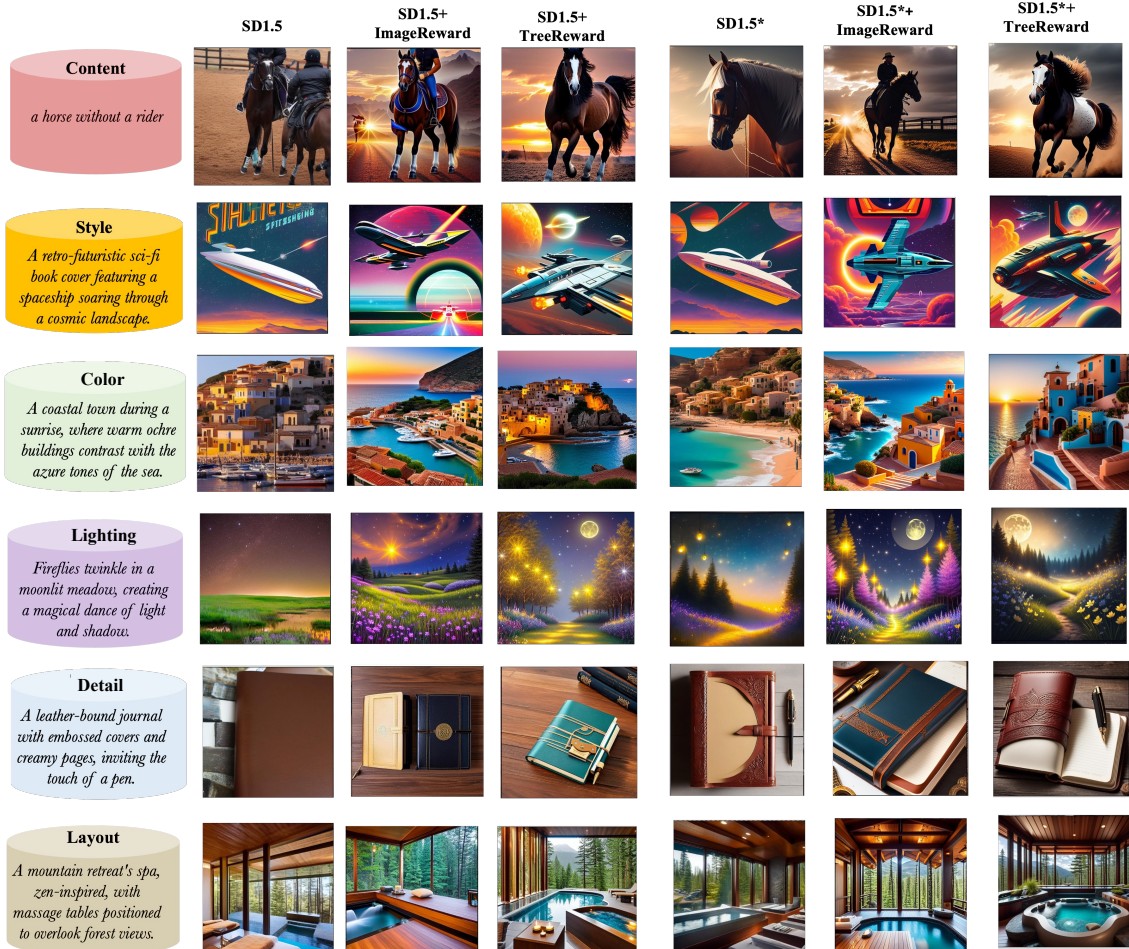

Figure 5: Visual comparison of each evaluation dimension. Among them, * means that the model has been fine-tuned in JourneyDB. TreeReward has achieved excellent results in terms of Content, Style, Color, Lighting, Detail, and Layout.

and aesthetic quality, respectively. The results are presented in Tab. 2. It demonstrates that our method outperforms the baseline model in both semantic alignment and aesthetic quality and also surpasses the performance of ImageReward. For instance, on SD1.5, our method achieved a 2% improvement in semantic alignment compared to ImageReward, along with a 1.6% enhancement in aesthetic quality. These results indicate the superiority of our method in generating images that are not only visually appealing but also semantically aligned. It is worth emphasizing that the score achieved by the SD1.5 model with fine-tuning using journey-db data is higher than that of the SDXL model. This observation underscores the significance of utilizing high-quality fine-tuned data to enhance

model performance. However, it is important to note that these metrics only provide a general overview. To obtain more detailed insights, we further conducted fine-grained human evaluation to validate the superiority of our method in generating high-quality images across diverse dimensions. Specifically, we take the original SD1.5 and JourneyDB-tuned SD1.5 as our baselines. Then, we evaluated the performance of the models optimized by ImageReward and TreeReward, respectively. Concretely, human raters were tasked with scoring the images generated by these two models ranging from -5 to 5 across various dimensions against the baseline models. The results of the fine-grained evaluation are presented in Tab.3.

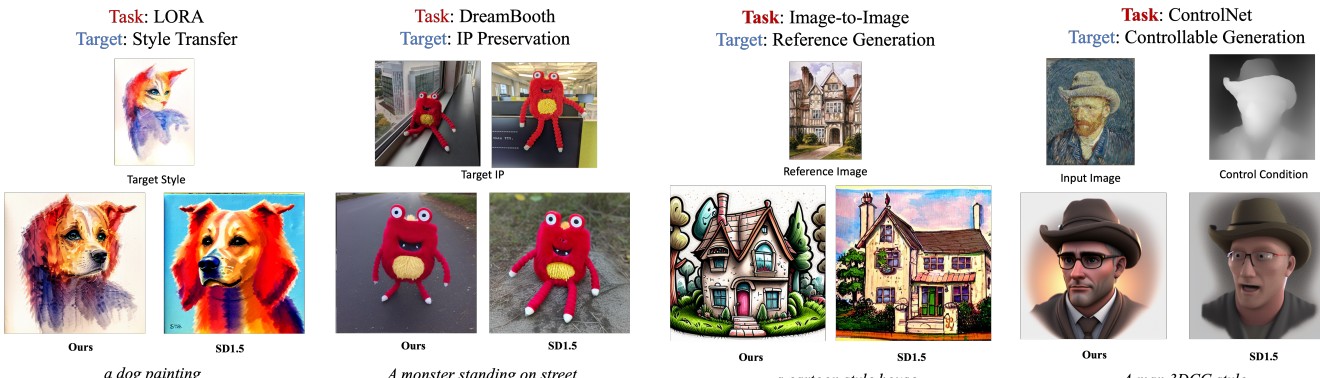

**Figure 6: Comparison of visual results on the downstream task with SD1.5 and SD1.5 optimized by our method.**

**Figure 7: The user study to validate feedback data scaling up, tree-structured reward feedback learning. The result is evaluated by 10 annotators on the generation of 100 prompts.**

**Table 4: 'IR': ImageReward. 'TR': TreeReward.**

| Setting | Data Volume | Structure | Reward Nums |
|---|---|---|---|
| ImageReward | 137K | IR | single |
| Data-Scale | 2.2M | IR | single |
| Decouple | 2.2M | IR | multiple |
| TreeReward (Ours) | 2.2M | TR | single |

It is evident that our TreeReward feedback learning approach significantly outperforms the ImageReward across all the dimensions. Remarkably, our TreeReward approach demonstrates a notable enhancement of 1.5 points in the 'Color' dimension when compared to the ImageReward method on SD1.5. Moreover, when applied on JourneyDB-tuned SD1.5, our TreeReward approach showcases a significant improvement of 2.2 points in the 'Detail' dimension, surpassing the ImageReward model by a substantial margin. Such comparison is also visualized in Fig.1, clearly demonstrating that our method outperforms ImageReward across all dimensions.

## 6.3 Ablation Study

We have conducted a series of ablation experiments to showcase the key contributions of our method, specifically the fine-grained feedback data scaling and the design and tree-structured reward fine-tuning. These experiments encompass several settings: (i) **"Data-Scale"**: We employ the same reward model as ImageReward but utilize our collected feedback data for training the reward model without distinguishing the different fine-grained dimensions. (ii) **"Decouple"**: Instead of training a single reward model, we train separate reward models for each fine-grained dimension using our feedback data and utilize these models for preference fine-tuning simultaneously. (iii) **"ImageReward"**: Preference fine-tuning using the reward model provided by ImageReward. (iv) **"TreeReward"**: Preference fine-tuning using the reward model provided by our

TreeReward approach. The detailed comparison between these settings is presented in Tab.4. As illustrated in Fig.7, incorporating more feedback data significantly enhances the performance of preference fine-tuning, resulting in an impressive increase of **18%** compared to the naive ImageReward approach. This finding underscores the importance of gathering a larger quantity of high-quality feedback data, even without considering fine-grained distinctiveness. Building upon this, the decoupling of the reward model for different fine-grained dimensions leads to a further improvement (44% vs 32%). This demonstrates the necessity of the decoupled reward model design, which effectively eliminates potential conflicts in preference tuning as analyzed in [27]. However, training multiple reward models not only results in memory inefficiency but also achieves sub-optimal multiple reward fine-tuning. In comparison, our TreeReward approach leverages fine-grained feedback data hierarchically and rewards in an adaptive manner, offering greater flexibility and delivering superior performance compared to naive fine-tuning with multiple reward models (54% vs 35%). By incorporating these two improvements, our method ultimately achieves a **32%** increase in user preference compared to ImageReward, highlighting the significant advantages of our approach.

## 6.4 Generalization Study

We conduct an extensive study to evaluate the generalization potential of our method in adapting to various downstream tasks, such as LORA [10], DreamBooth [24], Image-to-Image[16], and ControlNet [34]. As illustrated in Fig.6, our model showcases remarkable compacity in style learning, IP preservation, reference generation, and controllable generation.

## 7 CONCLUSION

We propose TreeReward, an effective method to boost the diffusion model across various fine-grained dimensions. It includes an efficient "AI + Expert" fine-grained feedback data construction pipeline, and a tree-structured reward model to achieve fine-grained, multi-dimension, and adaptive reward feedback learning. Extensive experiments on both SD1.5 and SDXL models demonstrate the superiority of our method in both boosting the general quality and fine-grained generation and at the same exhibiting excellent generalization.

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
