# OpenReview forum: "TreeReward: Improve Diffusion Model via Tree-Structured Feedback Learning"
_acmmm.org/ACMMM/2024/Conference — MM2024 Poster_

### Official Review · Reviewer_KJCV · 2024-04-28

**Rating:** 3
**Confidence:** 3

**Summary:**

This paper presents TreeReward, a novel multi-dimensional, fine-grained, and adaptive feedback learning framework that aims to improve both the semantic and aesthetic aspects of diffusion models. Firstly, this framework design an efficient feedback data construction pipeline in an "AI + Expert" fashion, yielding about 2.2M high-quality feedback dataset encompassing six fine-grained dimensions. Then, by introducing a tree-structure reward model, this framework improves model's performance of feedback learning. The extensive experiments demonstrate their method's effectiveness in enhancing the general and fine-grained generation performance and the generalizability of downstream
tasks.

**Strengths:**

1. This work designs an efficient feedback data curation pipeline that combines automated feedback generation and manual annotation to efficiently collect fine-grained feedback data on various aspects such as style, content, light, structure, layout, and detail, finally yielding about 2.2M feedback datase which is the largest dataset in the field.

2. This work introduces an innovative feedback learning method based on a tree-structure reward, TreeReward, that enables multi-dimensional, fine-grained, and adaptive preference feedback learning, resulting in more comprehensive and effective reward tuning.


3. Extensive experiments on both SD1.5 and SDXL validate the effectiveness of our method, showcasing superior performance when compared to the state-of-the-art reward tuning method. Additionally, the experiment on the downstream tasks further validates the efficacy and generalization of our method.

**Limitations:**

1. Is the impact of manually designing trigger words on the final model results significant, as mentioned in line 404 of the paper ?

2. In the paper, a large amount of aesthetic difference data was generated. How is this aesthetic difference defined ? Is it solely based on personal perception ? In figure3, how to define "preferred" and "unpreferred"?

3. In line 489, is it $r_{\theta}^{i}$ and $D_{j}$ ?
    In line 571, is it "and $\lambda$ is set to"?

4. I believe the sensitivity of the model's performance to lambda needs to be discussed.

5. I really like this data construction method, but I still have many questions about how to judge the quality of the image in this work. In the comparison results in Figure 5, I often feel that ImageReward is not necessarily worse than TreeReward. How can such subjective content be formally defined?

**Suitability:**

3

---

### Official Review · Reviewer_NA3Z · 2024-05-14

**Rating:** 4
**Confidence:** 2

**Summary:**

This work proposed a reward learning model for enhancing text-to-image generation performance. Fine-gained datasets are collected considering different dimensions of aesthetics and semantics. Then the adaptive reward learning to proposed for diffusion-based text-image generation models.

**Strengths:**

The results show the advantage of the treereward compared to other reward learning approaches. The collected datasets would help the community a lot if they are released. The effectiveness is shown with SD1.5 and SDXL models.

**Limitations:**

One concern is that authors ought to compare different reward learning methods more fairly, with the same volume of data. Otherwise, it is hard to say which part, the data or the reward learning, benefits more in the final performance.
It is also interesting to demonstrate how much more training overhead induced by the reward learning.
Regarding the claim "By combining the fine-grained rewards from all nodes, our model can adaptively focus on the reward dimensions that have not been well optimized yet,", I didn't see how equation 8 makes this possible. It seems like any leaf node getting more reward would be further rewarded by more chance. Does it cause any dominance  of some leaf nodes over others?

**Suitability:**

3

---

### Official Review · Reviewer_j3ph · 2024-05-23

**Rating:** 4
**Confidence:** 2

**Summary:**

In this work, the authors propose a method for text-to-image generation constrained by human feedback, following the recent success of similar training paradigms in the field of text generation. The manuscript presents two main issues in current solutions: that the current datasets are limited in sample size and that current feedback learning methods are crude. In order to tackle both problems, a semi-automated pipeline is proposed to assemble a new dataset with pair annotations much larger than existing datasets. By constructing their proposed dataset in 6 fine-grained dimensions, the authors then suggest a way to provide an adaptive preference of the different categories in a hierarchical fashion. A tree structure is essentially implemented in order to learn the most important preference when fine-tuning under a particular text prompt. Experiments and ablations on different versions of the Stable Diffusion model are performed in order to motivate the design and showcase its usefulness.

**Strengths:**

- The proposed dataset is a large step forward in terms of data scale compared to existing alternatives. In particular, the semi-automated strategy allows for easy extension of the dataset even further. It is also the only dataset to consider different types of preferences on the feedback type, which the authors refer to as dimensions. The annotation strategy is also a clever way of using MLLMs.

- TreeReward presents a simple way to separate the major feedback classes (semantics and aesthetics) in a learnable and dynamic fashion. This ensures that small modifications are more accurately reflected in the images produced by the fine-tuning procedure. The rewards create a sort of hierarchy in this sense, albeit not a very deep one. In some qualitative cases, TreeReward seems to produce much higher-fidelity samples.

**Limitations:**

- The experimental results of the proposed method are somewhat shaky. The results with TreeReward do not provide a major improvement compared to ImageReward, especially when considering SDXL (Table 1). Even when looking at qualitative examples, although both generated results are correct, there is no clear reason why one would prefer TreeReward-generated ones. A similar observation holds when looking at Table 3, where if one considers a rating scale of cardinality 10 (i.e., from -5 to 5), using ImageReward does not seem to provide much worse results on average. Related to the last statement, it would be beneficial to have the deviation of the average human rating in Table 3, especially since the number of human evaluators is not given. Qualitatively, while the benefits of TreeReward are visible on SD1.5, as mentioned previously, with SDXL, these improvements do not necessarily seem much better than the ImageReward counterpart. Indeed, in the main paper, the authors only present a visual comparison of each evaluation dimension. with SD1.5 and not with SDXL.

Minor:
- Some of the strategies for the dataset creation seem somewhat arbitrary. For example, the choice of having the diffusion model stop at different timesteps in order to generate positive-negative pairs is strange, as before reaching the final timestep, one is generating from the same mode, so the pair might not even be that different. Another aspect that deserves more detail should be presented in the vocabulary for the contrastive sample generation procedure since it has to be somewhat varied. As a minor point regarding this issue, shouldn't the timestep sequence in Fig. 2 be x_T -> x_{t2} -> x_{t1} -> x_0

- The caption of Fig. 1, line 82 seems to contain a mistake "three two components".

**Suitability:**

3

---

### Official Review · Reviewer_mxRR · 2024-05-31

**Rating:** 5
**Confidence:** 2

**Summary:**

The paper presents TreeReward, an innovative feedback-learning framework designed to enhance diffusion models using a tree-structured reward system. It addresses challenges such as insufficient data quantity and coarse feedback learning. By developing a comprehensive feedback data construction pipeline and a hierarchical reward model, TreeReward aims to improve several aspects, including Content, Style, Color, Lighting, Detail, and Layout. Extensive experiments demonstrate significant improvements over existing models, highlighting the framework's potential in various downstream tasks.

**Strengths:**

1. The authors have developed an efficient feedback data construction pipeline that combines automatic data generation with manual annotation. This approach has resulted in a dataset of 2.2 million high-quality feedback pairs, the largest in the field, which is a notable achievement.

2. The paper presents extensive experimental results on Stable Diffusion v1.5 and Stable Diffusion XL. The comparisons with the state-of-the-art ImageReward model are thorough and clearly demonstrate the superiority of TreeReward in both semantic alignment and aesthetic quality.

3. The proposed method includes more dimensions (=6) to evaluate the quality of generated images. By combining this with the tree-structured method, the performance of diffusion model can be effectively improved.

**Limitations:**

1. The main limitation of this paper is the manually designed tree structure used in the method. The proposed aspects are coarse-grained, which might limit the model's ability to capture finer nuances in feedback data.

2. Although the authors address the issue of data scalability by combining automated and manual feedback data generation, the process of manually annotating large datasets is still labor-intensive and costly. The reliance on such a large dataset might pose challenges for replication and scalability in different contexts or with different datasets.

3. The paper could provide a more balanced view by discussing the limitations and potential drawbacks of the proposed approach in greater detail. Acknowledging these aspects would offer a more comprehensive perspective and guide future research directions.

**Suitability:**

3

---

### Meta-Review · Area_Chair_fqC5 · 2024-07-04

**Recommendation:** Accept (Poster)
**Confidence:** 4

**Metareview:**

The manuscript has been reviewed by four experienced reviewers, all of whom, in the final rating, gave positive scores, indicating the manuscript indeed meets the bar of MM.

The AC therefore agrees with the consensus and recommends acceptance.

Please, however, do account for the comments in the final version.